# Tight Focusing of Circular Partially Coherent Radially Polarized Circular Airy Vortex Beam

**Zhihao Wan [1,2], Haifeng Wang [1,*], Cheng Huang [2], Zhimin He [2], Jun Zeng [2], Fuchang Chen [2], Chaoqun Yu [2], Yan Li [2], Huanting Chen [2], Jixiong Pu [2,3] and Huichuan Lin [2,*]**

1 China Academy of Engineering Physics, Institute of Fluid Physics, Mianyang 621900, China; lcj0198@mnnu.edu.cn

2 Key Laboratory of Light Field Manipulation and System Integration Applications in Fujian Province, College of Physics and Information Engineering, Minnan Normal University, Zhangzhou 363000, China; hc0223@mnnu.edu.cn (C.H.); hzm0191@mnnu.edu.cn (Z.H.); zj1838@mnnu.edu.cn (J.Z.); cfc1614@mnnu.edu.cn (F.C.); ycq1650@mnnu.edu.cn (C.Y.); ly1734@mnnu.edu.cn (Y.L.); cht1521@mnnu.edu.cn (H.C.); jixiong@hqu.edu.cn (J.P.)

3 Fujian Provincial Key Laboratory of Light Propagation and Transformation, College of Information Science & Engineering, Huaqiao University, Xiamen 361021, China

* Correspondence: hfwang@caep.cn (H.W.); lhc1810@mnnu.edu.cn (H.L.); Tel.: +86-18876330655 (H.L.)

**Abstract:** The tight focusing properties of circular partially coherent radially polarized circular Airy vortex beams (CPCRPCAVBs) are theoretically studied in this paper. After deriving the cross-spectral density matrix of CPCRPCAVBs in the focal region of a high-NA objective, numerical calculations were performed to indicate the influence of the topological charge of the vortex phase on intensity distribution, degree of coherence and degree of polarization of the tightly focused beam. An intensity profile along the propagation axis shows that a super-length optical needle (~15 λ) can be obtained with a topological charge of 1, and a super-length dark channel (~15 λ) is observed with a topological charge of 2 or 3. In the focal plane, the rise in the number of topological charge does not distort the shapes of the coherence distribution pattern and the polarization distribution pattern, but enlarges their sizes.

**Keywords:** tight focusing; radially polarized; circular partially coherent; circular Airy beam; topological charge; vortex phase

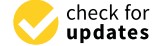



## 1. Introduction

Over the past decades, partially coherent beams with controllable spatial coherence have drawn considerable attention due to their extensive applications in optical communication, material thermal processing, particle trapping and so on [1–10]. Partially coherent beams with vortex phase have orbital angular momentum and exhibit unique correlation singularities [11–14]. Recent studies have shown that partially coherent beams with vortex phase have the advantage of reducing turbulence-induced scintillation [15,16]. Therefore, research into the generation and propagation properties of partially coherent vortex beams has attracted more and more attention. Due to the potential value of their auto-focusing property in optical micro-manipulation and biomedical applications, nonlinear beams such as circular Airy beams have been widely studied for years [17–21]. Extending the study from coherent nonlinear beams to partially coherent beams, Tong Li and his collaborators introduced the partially coherent radially polarized circular Airy beam, which not only exhibits auto-focusing capability but also creates an optical potential well with adjustable depth [22]. Santarsiero and his collaborators synthesized a new class of partially coherent light sources recently, namely a circular partially coherent light source [23]. Ding et al. demonstrated the self-focusing property of such beams on propagation in oceanic turbulence [24]. For a partially coherent radially polarized circular Airy beam with the spatial

coherence of circular partial coherence and a vortex phase, it will be of interest to study its propagation properties.

Using a laser beam tightly focused with a high-numerical-aperture (NA) objective, sub-wavelength focal spots or focal holes in the focal region can be obtained. Specifically, by modulating the amplitude, the polarization, the phase or the spatial coherence of the incident laser beam appropriately, sub-wavelength focal spots of desired shapes in the focal region can be achieved [25–30]. These designated sub-wavelength focal patterns are widely used in optical data storage, microscopy, particle beam trapping and material processing. Therefore, tight focusing of different kinds of laser beams, such as amplitude-modulated beams, phase-modulated beams, polarization-modulated beams and coherence-modulated beams, has attracted a great deal of interest for years [30]. As mentioned above, a circular Airy beam with the spatial coherence of circular partial coherence is an interesting phase-modulated beam, whose circular partial coherence is a special structure of spatial coherence. As a radially polarized beam is tightly focused, the smallest focal spot can be obtained [30]. Therefore, with regard to the combination of circular partial coherence modulation, circular Airy vortex phase modulation, radial polarization modulation and its tight focusing with a high-NA objective, we will characterize tightly focused circular partially coherent radially polarized circular Airy vortex beams (CPCRPCAVBs) in the focal region based on the Richards–Wolf vector diffraction theory.

## 2. Theoretical Analysis

In the source plane z = 0, the electric field of a radially polarized circular Airy beam with vortex phase can be expressed as [21]

$$E(r, \varphi, 0) = C_0 Ai\left(\frac{r_0 - r}{w}\right) \exp\left(a\frac{r_0 - r}{w}\right) \exp(im\theta)\left[\cos(\varphi)e_x + \sin(\varphi)e_y\right] \quad (1)$$

where $C_0$ is the normalization constant, $Ai$ is the Airy function, $w$ is the scaling parameter, $m$ is the topological charge of the vortex phase, $a$ is the decaying factor and $r_0$ is the radius of the first ring of the radially polarized circular Airy beam in the source plane. $(r, \varphi)$ in Equation (1) are the polar coordinates in the source plane. Referring to the expression of a partially coherent radially polarized circular Airy beam in reference [21], the electric cross-spectral density matrix of a CPCRPCAB with an $m$-order concentric vortex can be described as

$$W(r_1, \varphi_1, r_2, \varphi_2, 0) = C_0{}^2 Ai\left(\frac{r_0-r_1}{w}\right) Ai\left(\frac{r_0-r_2}{w}\right) \exp\left(a\frac{r_0-r_1}{w} + a\frac{r_0-r_2}{w}\right)$$
$$\times \exp[im(\varphi_1 - \varphi_2)]\sin c\left(\frac{r_2{}^2-r_1{}^2}{\delta}\right)\begin{bmatrix} \cos(\varphi_1)\cos(\varphi_2) & \cos(\varphi_1)\sin(\varphi_2) \\ \sin(\varphi_1)\cos(\varphi_2) & \sin(\varphi_1)\sin(\varphi_2) \end{bmatrix} \quad (2)$$

In Equation (2), $\sin c(x) = \frac{\sin(\pi x)}{\pi x}$ is the spatial coherence of the CPCRPCAVB, and the $\delta$ is the initial coherent length. For a laser beam tightly focused with a high-NA objective (NA > 0.7), the paraxial approximation is no longer applicable and so is the Kirchhoff integral theorem. The vectorial Debye theory should be employed to solve the cross-spectral density matrix of CPCRPCAVBs in the focal region of the high-NA objective.

The coordinate system is shown in Figure 1. According to the vectorial Debye theory, the second-order correlation properties of a CPCRPCAVB near the focal region of a high-NA objective can be characterized by a 3 × 3 electric cross-spectral density matrix $W(\rho_1, \theta_1, \rho_2, \theta_2, z)$. The elements of the 3 × 3 matrix are given by [31]

$$W(\rho_1, \psi_1, \rho_2, \psi_2, 0) = \left(\frac{C_0{}^2}{\lambda^2}\right)\int_0^\alpha \int_0^\alpha \int_0^{2\pi} \int_0^{2\pi} \sin c\left(\frac{f^2\sin^2\theta_2 - f^2\sin^2\theta_1}{\delta^2}\right)$$
$$\times P(\varphi_1, \theta_1)P(\varphi_2, \theta_2)K_i(\varphi_1, \theta_1)K_j^T(\varphi_2, \theta_2)\sqrt{\cos\theta_1}\sqrt{\cos\theta_2}s$$
$$\times \exp[-ikz\cos\theta_1 - ik\rho_1\sin\theta_1\cos(\psi_1 - \varphi_1)]in\theta_1\sin\theta_2$$
$$\times \exp[ikz\cos\theta_2 + ik\rho_2\sin\theta_2\cos(\psi_2 - \varphi_2)]d\varphi_1 d\varphi_2 d\theta_1 d\theta_2, (i, j = x, y, z) \quad (3)$$

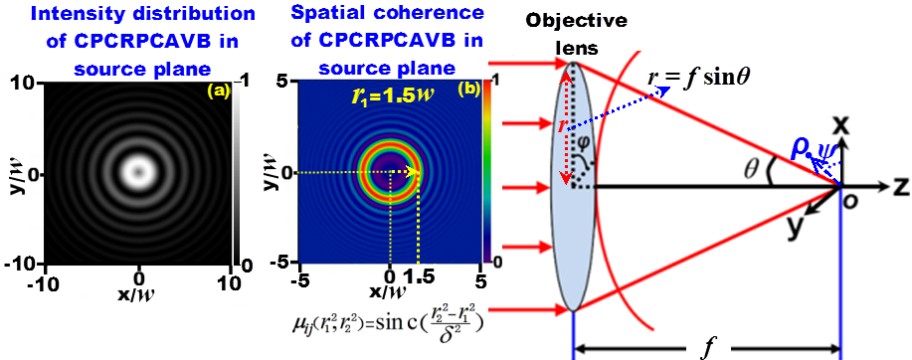

**Figure 1.** Tight focusing system. Inset (**a**) is the intensity distribution of the incident CPCRPCAVB, and (**b**) is the initial coherence of the incident CPCRPCAVB. The calculation parameters for the insets chosen are $\lambda$ = 633 nm, $w$ = 5 mm, $a$ = 0.1, $r_0$ = 1 mm and $\delta$ = 0.6 $w$.

In Equation (3), by replacing $r$ with $f \sin(\theta)$, the pupil apodization function of the CPCRPCAVB at the exit pupil of the high-NA objective can be expressed as

$$P(\varphi, \theta) = Ai\left(\frac{r_0 - f \sin\theta_1}{w}\right) \exp\left(a\frac{r_0 - f \sin\theta_1}{w}\right) \exp(im\varphi) \tag{4}$$

In Equation (3),

$$K(\varphi, \theta) = \begin{bmatrix} \cos\theta + \sin^2\varphi(1 - \cos\theta)e_x \\ \cos\varphi \sin\varphi(\cos\theta - 1)e_y \\ \cos\varphi \sin\theta e_z \end{bmatrix} \tag{5}$$

and the superscript T denotes the transposition of the matrix of, and thus $K^T(\varphi, \theta)$ is a $1 \times 3$ matrix. Through the same tedious integration operations, the elements of the $3 \times 3$ electric cross-spectral density matrix $W(\rho_1, \theta_1, \rho_2, \theta_2, z)$ of the CPCRPCAVB in the focal region of a high-NA objective can be obtained as follows:

$$
\begin{aligned}
W(\rho_1, \psi_1, \rho_2, \psi_2, 0) &= \left(\frac{4\pi^2 C_0^2}{\lambda^2}\right) \int_0^\alpha \int_0^\alpha \exp[ikz(\cos\theta_2 - \cos\theta_1)] \\
&\times \sqrt{\cos\theta_1}\sqrt{\cos\theta_2} \sin\theta_1 \sin\theta_2 Ai\left(\frac{r_0 - f\sin\theta_1}{w}\right) Ai\left(\frac{r_0 - f\sin\theta_2}{w}\right) \\
&\times \exp\left(a\frac{r_0 - f\sin\theta_1}{w} + a\frac{r_0 - f\sin\theta_2}{w}\right) \sin c\left(\frac{f^2\sin^2\theta_2 - f^2\sin^2\theta_1}{\delta^2}\right) \\
&\times M_{ij}(\rho_1, \rho_2, \psi_1, \psi_2, \theta_1, \theta_2) d\theta_1 d\theta_2, \quad (i,j = x, y, z)
\end{aligned} \tag{6}
$$

In Equation (6), $M_{ij}(\rho_1, \rho_2, \psi_1, \psi_2, \theta_1, \theta_2)$ for the elements of $W(\rho_1, \theta_1, \rho_2, \theta_2, z)$ are

$$
\begin{aligned}
M_{xx}(\rho_1, \rho_2, \psi_1, \psi_2, \theta_1, \theta_2) &= [\exp(i(m+1)\psi_1)\frac{1}{i^{m+1}}J_{m+1}(k\rho_1\sin\theta_1) \\
&+ i^{1-m}\exp(i(m-1)\psi_1)J_{m-1}(k\rho_1\sin\theta_1)]\cos\theta_1\cos\theta_2 \\
&\times [\exp(i(1-m)\psi_2)\frac{(-1)^{m-1}}{i^{m-1}}J_{m-1}(k\rho_2\sin\theta_2) \\
&+ \frac{(-1)^{m+1}}{i^{m+1}}\exp(-i(m+1)\psi_2)J_{m+1}(k\rho_2\sin\theta_2)]
\end{aligned} \tag{7}
$$

$$
\begin{aligned}
M_{yy}(\rho_1, \rho_2, \psi_1, \psi_2, \theta_1, \theta_2) &= [\exp(i(m-1)\psi_1)i^{1-m}J_{m-1}(k\rho_1\sin\theta_1) \\
&- \frac{1}{i^{(m+1)}}\exp(i(m+1)\psi_1)J_{m+1}(k\rho_1\sin\theta_1)]\cos\theta_1\cos\theta_2 \\
&\times [\exp(i(1-m)\psi_2)\frac{(-1)^{m-1}}{i^{m-1}}J_{m-1}(k\rho_2\sin\theta_2) \\
&- \frac{(-1)^{m+1}}{i^{m+1}}\exp(-i(m+1)\psi_2)J_{m+1}(k\rho_2\sin\theta_2)]
\end{aligned} \tag{8}
$$

$$
\begin{aligned}
M_{zz}(\rho_1, \rho_2, \psi_1, \psi_2, \theta_1, \theta_2) &= \sin\theta_1 \sin\theta_2 \exp[im(\psi_1 - \psi_2)] \\
&\times J_m(k\rho_1\sin\theta_1)J_m(k\rho_2\sin\theta_2)
\end{aligned} \tag{9}
$$

$$
\begin{aligned}
M_{xy}(\rho_1,\rho_2,\psi_1,\psi_2,\theta_1,\theta_2) &= [\exp(i(m+1)\psi_1)\tfrac{1}{i^{1+m}}J_{m+1}(k\rho_1\sin\theta_1) \\
&+ \tfrac{1}{i^{(m-1)}}\exp(i(m-1)\psi_1)J_{m-1}(k\rho_1\sin\theta_1)]\cos\theta_1\cos\theta_2 \\
&\times [\exp(i(1-m)\psi_2)\tfrac{(-1)^{m-1}}{i^{m-1}}J_{m-1}(k\rho_2\sin\theta_2) \\
&- \tfrac{(-1)^{m+1}}{i^{m+1}}\exp(-i(m+1)\psi_2)J_{m+1}(k\rho_2\sin\theta_2)]
\end{aligned}
\tag{10}
$$

$$
\begin{aligned}
M_{xz}(\rho_1,\rho_2,\psi_1,\psi_2,\theta_1,\theta_2) &= [\exp(i(m+1)\psi_1)\tfrac{1}{i^{1+m}}J_{m+1}(k\rho_1\sin\theta_1) \\
&+ \tfrac{1}{i^{(m-1)}}\exp(i(m-1)\psi_1)J_{m-1}(k\rho_1\sin\theta_1)]\cos\theta_1\sin\theta_2 \\
&\times \exp(-im\psi_2)J_m(k\rho_2\sin\theta_2)\tfrac{(-1)^m}{i^m}J_m(k\rho_2\sin\theta_2)
\end{aligned}
\tag{11}
$$

$$
\begin{aligned}
M_{yx}(\rho_1,\rho_2,\psi_1,\psi_2,\theta_1,\theta_2) &= [\exp(i(m+1)\psi_1)\tfrac{1}{i^{1+m}}J_{m+1}(k\rho_1\sin\theta_1) \\
&- \tfrac{1}{i^{(m-1)}}\exp(i(m-1)\psi_1)J_{m-1}(k\rho_1\sin\theta_1)]\cos\theta_1\cos\theta_2 \\
&\times [\exp(i(1-m)\psi_2)\tfrac{(-1)^{m-1}}{i^{m-1}}J_{m-1}(k\rho_2\sin\theta_2) \\
&+ \tfrac{(-1)^{m+1}}{i^{m+1}}\exp(-i(m+1)\psi_2)J_{m+1}(k\rho_2\sin\theta_2)]
\end{aligned}
\tag{12}
$$

$$
\begin{aligned}
M_{yz}(\rho_1,\rho_2,\psi_1,\psi_2,\theta_1,\theta_2) &= [\exp(i(m+1)\psi_1)\tfrac{1}{i^{1+m}}J_{m+1}(k\rho_1\sin\theta_1) \\
&- \tfrac{1}{i^{(m-1)}}\exp(i(m-1)\psi_1)J_{m-1}(k\rho_1\sin\theta_1)] \\
&\times \cos\theta_1\sin\theta_2\exp(-im\psi_2)\tfrac{(-1)^m}{i^m}J_m(k\rho_2\sin\theta_2)
\end{aligned}
\tag{13}
$$

$$
\begin{aligned}
M_{zx}(\rho_1,\rho_2,\psi_1,\psi_2,\theta_1,\theta_2) &= [\exp(-i(1-m)\psi_2)J_{m-1}(k\rho_2\sin\theta_2) \\
&+ \exp(-i(m+1)\psi_2)J_{m+1}(k\rho_2\sin\theta_2)] \\
&\times \sin\theta_1\cos\theta_2\exp(im\psi_1)J_m(k\rho_1\sin\theta_1)(-1)^m
\end{aligned}
\tag{14}
$$

$$
\begin{aligned}
M_{zy}(\rho_1,\rho_2,\psi_1,\psi_2,\theta_1,\theta_2) &= [\exp(i(1-m)\psi_2)J_{m-1}(k\rho_2\sin\theta_2) \\
&- \exp(-i(m+1)\psi_2)J_{m+1}(k\rho_2\sin\theta_2)] \\
&\times \sin\theta_1\cos\theta_2\exp(im\psi_1)J_m(k\rho_1\sin\theta_1)(-1)^m
\end{aligned}
\tag{15}
$$

In Equations (7)–(15), $J_n$ are the $n$-order Bessel functions of the first kind. Setting $\rho_1 = \rho_2$ and $\psi_1 = \psi_2$, the intensity of the $x$-polarization component, $y$-polarization component, $z$-polarization component and total intensity distribution of the CPCRPCAVB in the focal region can be expressed as:

$$
I_x(\rho_1,\rho_1,\psi_1,\psi_1,z) = W_{xx}(\rho_1,\rho_1,\psi_1,\psi_1,z)
\tag{16}
$$

$$
I_y(\rho_1,\rho_1,\psi_1,\psi_1,z) = W_{yy}(\rho_1,\rho_1,\psi_1,\psi_1,z)
\tag{17}
$$

$$
I_z(\rho_1,\rho_1,\psi_1,\psi_1,z) = W_{zz}(\rho_1,\rho_1,\psi_1,\psi_1,z)
\tag{18}
$$

$$
I_{total}(\rho_1,\rho_1,\psi_1,\psi_1,z) = I_x(\rho_1,\rho_1,\psi_1,\psi_1,z) + I_y(\rho_1,\rho_1,\psi_1,\psi_1,z) + I_z(\rho_1,\rho_1,\psi_1,\psi_1,z)
\tag{19}
$$

In addition, with the $3 \times 3$ matrix of

$$
W(\rho_1,\rho_2,\psi_1,\psi_2,z) = \begin{bmatrix} W_{xx}(\rho_1,\rho_2,\psi_1,\psi_2,z) & W_{xy}(\rho_1,\rho_2,\psi_1,\psi_2,z) & W_{xz}(\rho_1,\rho_2,\psi_1,\psi_2,z) \\ W_{yx}(\rho_1,\rho_2,\psi_1,\psi_2,z) & W_{yy}(\rho_1,\rho_2,\psi_1,\psi_2,z) & W_{yz}(\rho_1,\rho_2,\psi_1,\psi_2,z) \\ W_{zx}(\rho_1,\rho_2,\psi_1,\psi_2,z) & W_{zy}(\rho_1,\rho_2,\psi_1,\psi_2,z) & W_{zz}(\rho_1,\rho_2,\psi_1,\psi_2,z) \end{bmatrix}
\tag{20}
$$

the degree of coherence of the CPCRPCAVB in the focal region can also be obtained. The complex correlation coefficient at two points, $(\rho_1, \psi_1, z)$ and $(\rho_2, \psi_2, z)$, in the focal region is expressed as

$$
\mu_{ij}(\rho_1,\rho_2,\psi_1,\psi_2,z) = W_{ij}(\rho_1,\rho_2,\psi_1,\psi_2,z)/\sqrt{W_{ii}(\rho_1,\psi_1,z)W_{jj}(\rho_2,\psi_2,z)}
\tag{21}
$$

Similarly, with the electric cross-spectral density matrix $W(\rho_1, \theta_1, \rho_2, \theta_2, z)$, the degree of polarization of the tightly focused CPCRPCAVB in the focal region can be given by [32]

$$P(\rho, \psi, z) = \sqrt{\frac{3}{2} \left\{ \frac{I_x(\rho, \psi, z)^2 + I_y(\rho, \psi, z)^2 + I_z(\rho, \psi, z)^2}{\left[ I_x(\rho, \psi, z) + I_y(\rho, \psi, z) + I_z(\rho, \psi, z) \right]^2} - \frac{1}{3} \right\}} \quad (22)$$

The degree of polarization as shown in Equation (22) can be used to describe the depolarization of the tightly focused CPCRPCAB in the focal plane. The coefficient of the degree of polarization is limited to the interval of $0 \leq P(\rho, \psi, z) \leq 1$, in which the upper limit represents a completely polarized beam, the lower limit represents a completely unpolarized beam and a value between them indicates a partially polarized beam.

### 3. Results and Discussion

In order to indicate the tight focusing properties of the CPCRPCAVBs more explicitly, some numerical calculations were carried out based on the above equation. General-purpose mathematical computation software can be used for the numerical calculations, which can also be accomplished by custom programs written in Python or C language. In addition, the numerical calculations mentioned in this paper are not computationally intensive and can be done using common PCs. Firstly, intensity distributions of the $x$-polarization component, $y$-polarization component, $z$-polarization component and total intensity of the CPCRPCAVB with different topological charges in the focal plane are shown in Figure 2. The wavelength of the CPCRPCAVB is taken as $\lambda = 633$ nm, which is used extensively in the studies of laser beam propagation. The scaling parameter $w$ and the parameter $r_0$, which determines the radius of the first ring of the CPCRPCAVB in the initial plane, were taken as 5 mm and 1 mm, respectively. The decaying factor $a$ is a dimensionless parameter, which was taken as 0.1 in the numerical calculation. The initial coherence length $\delta$ was taken as $0.6 \, w$, and its corresponding initial coherence distribution of CPCRPCAVB is shown in inset (b) of Figure 1. Obviously, this value of the initial coherence length is very representative. The coherence distribution exhibits perfect coherence along any annulus that is concentric to the source center. In contrast, for two points at different distances from the center, the coherence is partial or has even vanished. It can be seen that when the topological charge, $m$, is equal to 1, the intensity distribution of the $x$-polarization component exhibits a central principal maximum. Along the direction of the $x$-axis, there are intensity minimums at both sides of the principal maximum. As the number of topological charge ($m$) rises to 2 or 3, the shape of the central principal maximum expands and a hollow appears in its center. The intensity of the $y$-polarization component exhibits a similar distribution to that of the $x$-polarization component, with the difference being that the direction of the pattern is turned by 90 degrees. The intensity distributions of the $z$-polarization with $m = 1$, $m = 2$ and $m = 3$ in the focal plane are illustrated in Figure 2g, h and i, respectively. They exhibit a Newtonian ring distribution with an intensity minimum or dark hollow in the center. With the increase in the topological charge, the hollow region gradually expands its size. The total intensity distributions of the CPCRPCAVB with $m = 1$, $m = 2$ and $m = 3$ in the focal plane are illustrated in Figure 2j, k and l, respectively. For the figure of mboxemphm = 1, the appearance of total intensity in the focus is like a Gaussian distribution, and the radial polarization and vortex phase do not introduce singularity in the focal plane. As the topological charge rises to $m = 2$ or $m = 3$, a hollow spot appears in the center of the intensity distribution. Comparing Figure 2k,l, it can be found that the size of the hollow spot increases with the increase in topological charge.

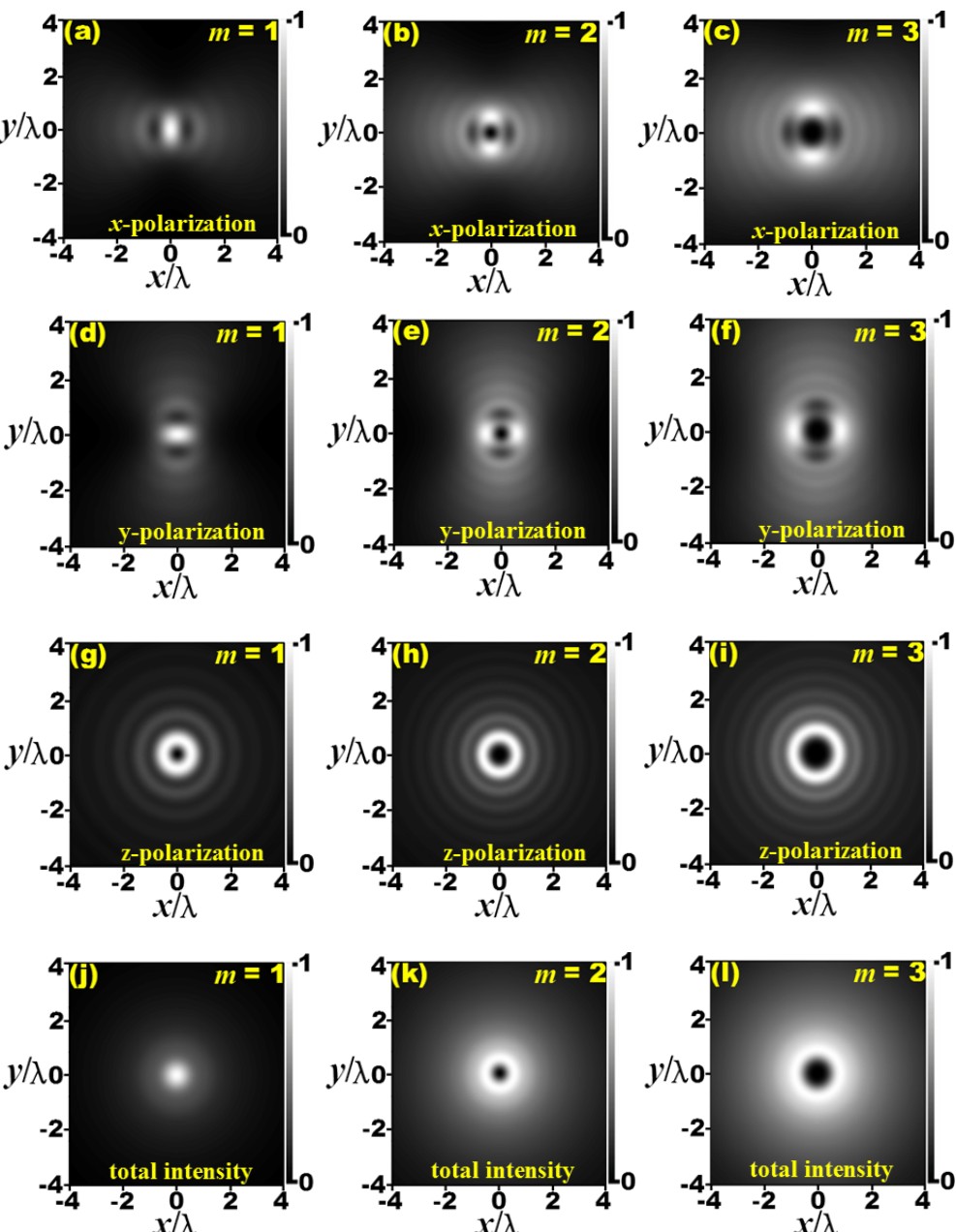

**Figure 2.** Intensity distributions of the CPCRPCAVB in the focal plane. (**a**–**c**) intensity distributions of the *x*-polarization component for different topological charge; (**d**–**f**) intensity distributions of the *y*-polarization component for different topological charge; (**g**–**i**) intensity distributions of the *z*-polarization component for different topological charge; (**j**–**l**) intensity distributions of the total intensity for different topological charge. The parameters for calculation chosen are λ = 633 nm, *w* = 5 mm, NA = 0.9, *f* = 1 cm, $C_0$ = 1, *a* = 0.1, $r_0$ = 1 mm and δ = 0.6 *w*.

The total intensity distributions of the tightly focused CPCRPCAVB in the *ρ*-*z* plane (i.e., propagation plane) near the focus are illustrated in Figure 3. One can find that when the topological charge of *m* is equal to 1, the intensity distribution of the CPCRPCAVB exhibits a needle-like shape along the optical axis, and the length of the homogeneous intensity along the optical axis is measured to be about 15λ. As the topological charge rises to *m* = 2 and *m* = 3, the needle pattern is divided into two halves by a dark channel on the optical axis, which is a non-diffracting focal hole surrounded by the regions of higher intensity in the radial direction, and the increase in topological charge leads to an enlargement in the radial dimension of the dark channel. This means that the radial dimension of the dark

channel can be controlled by adjusting the topological charge of the incident CPCRPCAVB. In a word, a super-length optical needle or dark channel can be obtained by adjusting the topological charge of a CPCRPCAVB tightly focused with a high-NA objective. Compared with other methods, the method proposed in this paper is relatively convenient in terms of obtaining the super-length optical needle and dark channel with a radial dimension of sub-wavelength scale, simply tightly focusing the CPCRPCAVB with a high-NA objective without any other auxiliary optical components. The optical needle and dark channel with a radial dimension of sub-wavelength scale may find potential applications in optical data storage, photolithography, super-resolution microscopy and particle trapping.

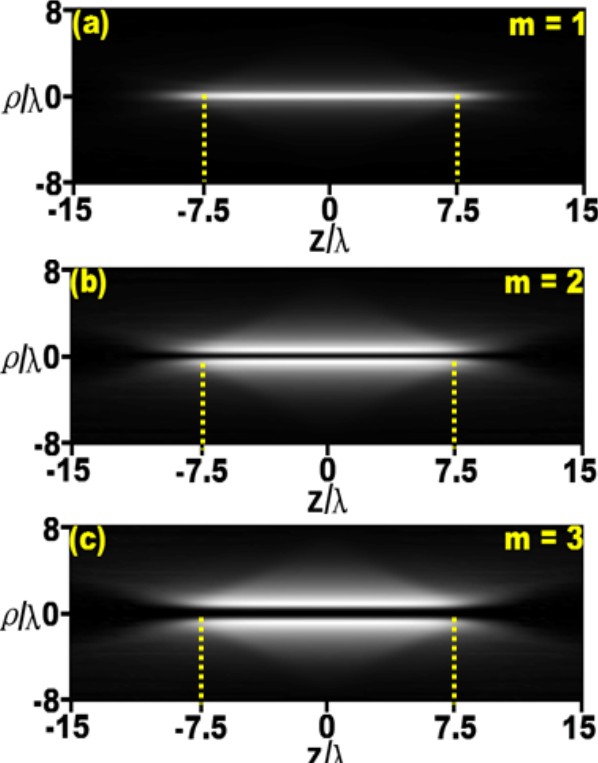

**Figure 3.** Total intensity distributions of the CPCRPCAVB in the $\rho$-$z$ plane near the focus. (**a**) topological charge is $m = 1$, (**b**) topological charge is $m = 2$, (**c**) topological charge is $m = 3$. The parameters for calculation chosen are $\lambda = 633$ nm, $w = 5$ mm, NA $= 0.9$, $f = 1$ cm, $C_0 = 1$, $a = 0.1$, $r_0 = 1$ mm and $\delta = 0.6\,w$.

The correlation between any two parallel electric field components near the focus can be characterized by $\left|\mu_{ii}(\rho_1, \rho_2, \theta_1, \theta_2, z), (i = x, y, z)\right|$, where $\rho$ and $\theta$ are the radius and angle in the polar coordinate system. For simplicity, setting the position of the first point $(\rho_1, \theta_1)$ in the geometrical focus of the high-NA objective (*i.e.*, $\rho_1 = 0$, $\theta_1 = 0$), the distributions of coherence for both parallel electric field components in the focal plane are calculated and shown in Figure 4. Figure 4a–c are the distributions of $\mu_{xx}$ for $m = 1$, $m = 2$ and $m = 3$, which all exhibit human-face-like patterns. The "nose" in the middle of each pattern is an intermediate principal maximum, and the "eyes" by the "nose" are two principal minimums symmetrically distributed along the x-axis on either side of the principal maximum. The distribution of $\mu_{yy}$ is similar to that of $\mu_{xx}$, except that its principal maximum and minimums are distributed along the y-axis. As shown in Figure 4g–i, the distributions of $\mu_{zz}$ are center-symmetric, exhibiting a wave-like circular oscillation from the central main pole outward. Comparing the distributions of $\mu_{xx}$, $\mu_{yy}$ and $\mu_{zz}$ with different topological charges, it can be found that the rise in topological charge of the vortex phase does not change the basic structure of the coherence distribution pattern, but increases its size.

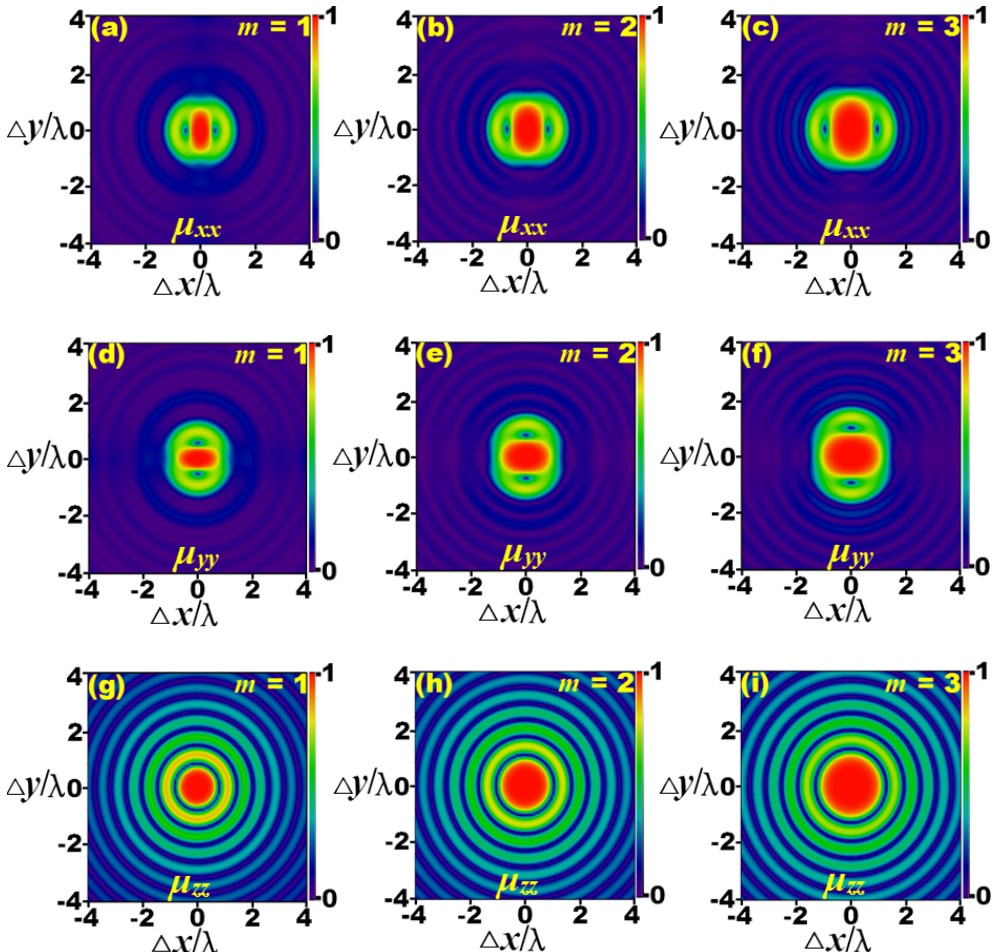

**Figure 4.** Distributions of coherence of the CPCRPCAVB with different topological charges in the focal plane. (**a**–**c**) coherence distributions of $\mu_{xx}$ for different topological charge; (**d**–**f**) coherence distributions of $\mu_{yy}$ for different topological charge; (**g**–**i**) coherence distributions of $\mu_{zz}$ for different topological charge. The parameters for calculation chosen are $\lambda$ = 633 nm, $w$ = 5 mm, NA = 0.9, $f$ = 1 cm, $C_0$ = 1, $a$ = 0.1, $r_0$ = 1 mm and $\delta$ = 0.6 $w$.

In order to learn about the polarization properties of the tightly focused CPCRPCAVB in the focal plane, the degree of polarization of the CPCRPCAVB with different topological charges is calculated, and the results are plotted in Figure 5. It can be seen that the polarization degree of the tightly focused CPCRPCAVB is not equal to 1 in most areas of the focal plane. The patterns take on an appearance of oscillation from the center outward along the radius. There is a cross shape surrounded by a circular ring in the center of each figure. Moreover, in the region outside the ring, the degree of polarization in the areas of the azimuth of the 0-degree and 90-degree directions is significantly higher than that of the azimuth of the 45-degree and 135-degree directions. By comparing the polarization distributions with different topological charges, it can be found that the patterns of the cross and the circular ring gradually expand outward along the radial direction with the increase in topological charge.

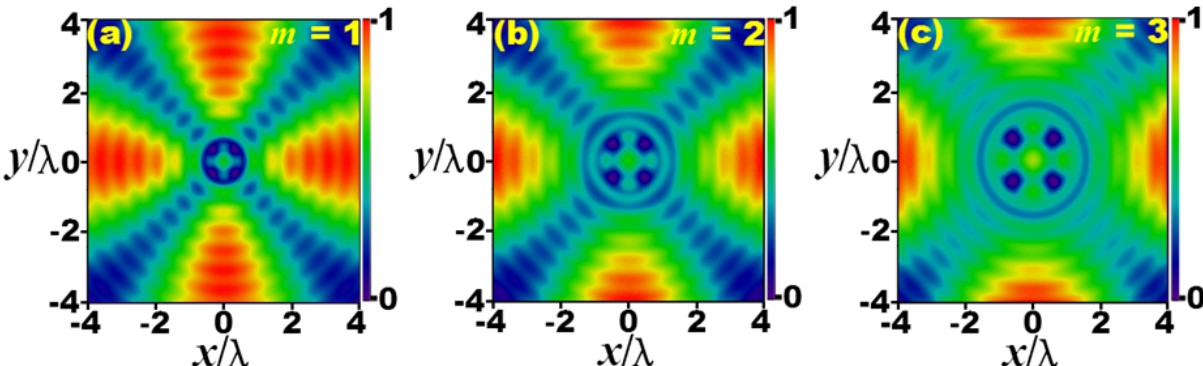

**Figure 5.** Degree of polarization of tightly focused CPCRPCAVB with different topological charges in the focal plane. (**a**) topological charge is $m = 1$, (**b**) topological charge is $m = 2$, (**c**) topological charge is $m = 3$. The parameters for calculation chosen are $\lambda = 633$ nm, $w = 5$ mm, NA = 0.9, $f = 1$ cm, $C_0 = 1$, $a = 0.1$, $r_0 = 1$ mm and $\delta = 0.6\,w$.

## 4. Conclusions

In conclusion, we have studied the tight focusing properties of circular partially coherent radially polarized circular Airy vortex beams (CPCRPCAVBs) in this paper. This study focuses on the effect of topological charge on their tight focusing properties, including intensity distribution, spatial coherence and degree of polarization. The results show that a super-length optical needle or a super-length dark channel can be obtained as the CPCRPCAVB is tightly focused. The super-length dark channel is a non-diffracting focal hole surrounded by regions of higher intensity in radial direction. By adjusting the topological charge of the vortex phase, the radial dimension of the dark channel can be controlled easily. It is simple and convenient to use this method to obtain a super-length optical needle and a super-length dark channel, and it may find application in optical data storage, photolithography, super-resolution microscopy and particle trapping. In the focal region, the rise in the number of topological charge does not distort the basic structures of the coherence distribution pattern and the polarization distribution pattern, but enlarges their sizes.

**Author Contributions:** Conceptualization, H.L.; methodology, H.W.; software, C.H. and F.C.; validation, J.Z., Y.L. and C.Y.; formal analysis, Z.H. and H.C.; investigation, Z.W.; resources, H.L. and H.W.; data curation, Z.W.; writing—original draft preparation, Z.W.; writing—review and editing, H.L. and H.W.; visualization, Z.W.; supervision, H.L. and J.P.; project administration, H.L.; funding acquisition, H.L. and H.W. All authors have read and agreed to the published version of the manuscript.

**Funding:** This research was funded by the National Natural Science Foundation of China (NSFC), grant numbers 61975072 and 12174173. This research was also funded by the Natural Science Foundation of Fujian province, grant numbers 2022H0023, 2022J02047 and 2022G02006.

**Institutional Review Board Statement:** Not applicable.

**Informed Consent Statement:** Not applicable.

**Data Availability Statement:** No new data were created.

**Conflicts of Interest:** No conflict of interest exits in this manuscript, and manuscript is approved by all authors.

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
