# Peer review of "Tight Focusing of Circular Partially Coherent Radially Polarized Circular Airy Vortex Beam"

_photonics, doi:10.3390/photonics10111279_

Round 1

Reviewer 1 Report

Comments and Suggestions for Authors

Author Response

Dear Reviewer,

Thank you very much for your comments. We have revised the manuscript, and would like to re-submit it for your consideration. We have addressed the comments raised by you, and the amendments are highlighted in red in the revised manuscript. Point by point responses to your comments are listed in the "Responses to the Reviewers comments".

I look forward to hearing from you soon.

With best wishes,

Yours sincerely,

Huichuan Lin

Reviewer 2 Report

Comments and Suggestions for Authors

Title: Tight focusing of circular partially coherent radially polarized circular Airy vortex beam

The theoretical part of the article is acceptable although some transitions between equations are confusing. However, the results section cannot be assessed since nothing is explained about the numerical simulations carried out. In that sense, the conclusions obtained are not supported by the results that are not evaluable.

[MAJOR COMMENTS]

[MAJOR COMMENT 1-line 125] “some numerical simulations were carried out…” Suddenly, without explaining anything in methodology, this expression appears “some numerical simulations were carried out…”. The results must be preceded by explaining the software, computers, simplifications, and models adopted to carry out the numerical simulations. These obtained results, without having explained it before, are not publishable nor does it have a scientific sound.

[MAJOR COMMENT 2-Figures Captions 1, 2, 3 and 4] Why have you chosen these parameters and not others? It is acceptable to indicate this in the legend of these figures, but they should also appear in the literature of the paper. Please include them in the body of the article as well. If we had chosen others, would the results change? Please justify your answer.

[MINOR COMMENTS/SUGGESTIONS]

[lines 56-58] Please, you must indicate the units (m, a, r0…) in parentheses.

[lines 72-74] The transition from Equation 3 to Equation 5 is confusing.

[line 75] matrix of…? Please, you must indicate it.

During the manuscript, I find equation, Equation, Eq. or nothing... Please, you must unify the criteria to denote the word equation in the same way throughout the manuscript.

ENGLISH STYLE

Moderate editing of English language required.

OVERALL RECOMENDATION

Reject.

Comments on the Quality of English Language

Moderate editing of English language required.

Author Response

(The authors gave the same response as above.)

Reviewer 3 Report

Comments and Suggestions for Authors

Abstract: In this paragraph I could not find the real objectives. It was not clear.

Introduction

Here again until now I do nor know the objective.

 Theoretical analysis 

in line 55, the equation needs to be defined in text before be showed and the referecce for this equation... Do this for all equations. In additcion, I believe you could use math equation editor to include euler notation. Some equation are very large and difficult to understand 

After all equations, it would be interesting a therical figure to ilustrate your goal.

Spreading and Wander of Asymmetric Bessel-Gaussian Beam 

“When asymmetric Bessel-Gaussian beam propagates in turbulent ocean” here one figure to ilustrate this beam.

Results and discussions 

“the oceanic turbulence effects on the spreading and wander of 200 asymmetric Bessel-Gaussian beam are first analyzed ” here one figure to ilustrate it

please, not use this notation Fig. 3 , use Figure 3.

Here is missing before the theorical experiment definition. So after that the reader can realize you results comes from. The figure 3 I not sure about structure constant, só the figure ir was mpossible to get idea.

Conclusion 

Here, It was impossible to agree with the manuscript. The paper do not exist any metrics, and the theorical experiment to give an idea the objectives and compare with figure 1,2,3 and 4.

I believe this manuscript needs to be more improving.

Comments on the Quality of English Language

That's fine

Author Response

(The authors gave the same response as above.)

Reviewer 4 Report

Comments and Suggestions for Authors

Dear Colleagues,

Congratulations on the idea you want to present with this manuscript. However, it would need some modifications.

First, some items in the Bibliography are more than five years old. It would be good to add newer works to the Bibliography, and obviously, you will need to redo the Introduction. Also, in the Introduction, it would be good to keep one way of referring to the bibliographical items and to put your proposed topic in the current context.

Section 2 is good in that it is broad, and the reader can follow the calculations, but simultaneously, the way of rendering the equations is heavy. It might be a good idea to make some compressions or a graphic diagram of the calculation steps.

In the Results and Discussion section, it needs to be clarified what your contribution and novelty are in applying the proposed method.

 Also, in the Conclusions, highlight the strengths of your proposed method.

Comments on the Quality of English Language

Minor editing of English language required.

Author Response

(The authors gave the same response as above.)

Reviewer 5 Report

Comments and Suggestions for Authors

The paper calls: "Tight focusing of circular partially coherent radially polarized circular Airy vortex beam" and concerned of theoretical calculations of spreading (focusing) of partly polarized radiation. The advantage of article is modern and actual topic of research with long references list (32 ones).

However i have some questions:

1. Where of experiment or it's just theoretical simulation?

2. Which software was used for such mathematical modeling?

Author Response

(The authors gave the same response as above.)

Round 2

Reviewer 1 Report

Comments and Suggestions for Authors

The authors have responded to most of the comments by adding a new paragraph, a few references, and an explanatory Figure 1 to the manuscript. The paper is now more accessible to a wider audience, including experts in related fields.

Reviewer 2 Report

Comments and Suggestions for Authors

Title: Tight focusing of circular partially coherent radially polarized circular Airy vortex beam

GENERAL COMMENT 2nd ROUND

The authors have justified their decisions based on my comments and those of other reviewers. When necessary, suggestions have been incorporated into the final version of the manuscript.

ENGLISH STYLE

Minor editing of English language required.

OVERALL RECOMENDATION

Accept in present form.

Comments on the Quality of English Language

Minor editing of English language required.

Reviewer 3 Report

Comments and Suggestions for Authors

Now is better. 

Comments on the Quality of English Language

I just like to read Figure instead of Fig. 

Reviewer 4 Report

Comments and Suggestions for Authors

Dear Colleagues,

I wish you success!

Thank you for considering my views.